# The Cellular Prion Protein Increases the Uptake and Toxicity of TDP-43 Fibrils

**DOI:** 10.3390/v13081625

**Published:** 2021-08-17

**Authors:** Carlo Scialò, Luigi Celauro, Marco Zattoni, Thanh Hoa Tran, Edoardo Bistaffa, Fabio Moda, Robert Kammerer, Emanuele Buratti, Giuseppe Legname

**Affiliations:** 1Laboratory of Prion Biology, Department of Neuroscience, Scuola Internazionale Superiore di Studi Avanzati (SISSA), 34136 Trieste, Italy; cscialo@sissa.it (C.S.); lcelauro@sissa.it (L.C.); mzattoni@sissa.it (M.Z.); hoa.tran@vnuk.edu.vn (T.H.T.); 2Unit of Neurology 5 and Neuropathology, Fondazione IRCCS Istituto Neurologico Carlo Besta, 20133 Milan, Italy; edoardo.bistaffa@istituto-besta.it (E.B.); Fabio.Moda@istituto-besta.it (F.M.); 3Institute of Immunology, Friedrich-Loeffler Institute, 17493 Riems, Germany; Robert.Kammerer@fli.de; 4Molecular Pathology Laboratory, International Centre for Genetic Engineering and Biotechnology (ICGEB), 34149 Trieste, Italy; buratti@icgeb.org

**Keywords:** PrP^C^, TDP-43, fibrils, uptake, toxicity

## Abstract

Cytoplasmic aggregation of the primarily nuclear TAR DNA-binding protein 43 (TDP-43) affects neurons in most amyotrophic lateral sclerosis (ALS) and approximately half of frontotemporal lobar degeneration (FTLD) cases. The cellular prion protein, PrP^C^, has been recognized as a common receptor and downstream effector of circulating neurotoxic species of several proteins involved in neurodegeneration. Here, capitalizing on our recently adapted TDP-43 real time quaking induced reaction, we set reproducible protocols to obtain standardized preparations of recombinant TDP-43 fibrils. We then exploited two different cellular systems (human SH-SY5Y and mouse N2a neuroblastoma cells) engineered to express low or high PrP^C^ levels to investigate the link between PrP^C^ expression on the cell surface and the internalization of TDP-43 fibrils. Fibril uptake was increased in cells overexpressing either human or mouse prion protein. Increased internalization was associated with detrimental consequences in all PrP-overexpressing cell lines but was milder in cells expressing the human form of the prion protein. As described for other amyloids, treatment with TDP-43 fibrils induced a reduction in the accumulation of the misfolded form of PrP^C^, PrP^Sc^, in cells chronically infected with prions. Our results expand the list of misfolded proteins whose uptake and detrimental effects are mediated by PrP^C^, which encompass almost all pathological amyloids involved in neurodegeneration.

## 1. Introduction

Cytoplasmic aggregated deposits of the nuclear TAR DNA-binding protein 43 (TDP-43) characterize neurons in most amyotrophic lateral sclerosis (ALS) and approximately half of frontotemporal lobar degeneration (FTLD) cases, now collectively called TDP-43 proteinopathies [1,2,3]. TDP-43 is a nucleo-cytoplasmic shuttling protein comprised of two RNA recognition motifs (RRMs), an N-terminal (NTD) and a C-terminal prion-like or low complexity domain (LCD) which mediates protein-protein interactions and its incorporation into stress granules [4,5,6,7,8,9]. TDP-43 LCD is crucially involved in disease since it is proteolytically cleaved and abnormally phosphorylated, leading to its cytoplasmic accumulation in complex with the full-length protein [3]. Misfolded forms of TDP-43 share some features with prions, as the ability to act as seeds of replication in vitro and in vivo [10,11,12,13] and the tendency to acquire different conformational states (i.e., strains) within affected tissues [14,15]. Hence, despite the lack of infectivity, as other neurodegeneration-associated proteins like amyloid-beta (Aβ), α-synuclein (α-syn) and tau, TDP-43 has been designated as a prion-like protein [3,16]. In the last decade several studies recognized the cellular prion protein, PrP^C^, as a common ligand and downstream effector of circulating neurotoxic species of several proteins involved in neurodegeneration, including the misfolded forms of the prion protein, PrP^Sc^, Aβ, α-syn and tau [17,18,19,20,21,22,23,24,25,26,27,28,29,30,31,32,33,34,35,36,37]. An indirect proof in support of PrP^C^ and amyloid interaction derived also from the various observations of reduced PrP^Sc^ accumulation in cells chronically infected with prions upon treatment with specific species of Aβ, α-syn or tau fibrils [30,34,38]. One of the hypotheses underlying this effect is that when PrP^C^ on the cell surface is coated by these fibrils, its rate of conversion into its pathological counterpart, PrP^Sc^, is reduced.

Here, capitalizing on our recently adapted TDP-43 real-time quaking induced reaction (RT-QuIC) [39] we aimed at investigating the involvement of PrP^C^ in the uptake of TDP-43 fibrils and its potential role in mediating at least part of their detrimental effects. We devised an in vitro assay to verify the interaction between TDP-43 LCD fibrils and recombinant PrP^C^ and subsequently tested TDP-43 fibril toxicity in two different cell lines, human SH-SY5Y and mouse neuroblastoma (N2a) cells, engineered to express low or high levels of PrP^C^. We verified by means of immunofluorescence the intracellular distribution of the internalized TDP-43 LCD fibrils in SH-SY5Y cells, observing that they induced the formation of cytoplasmatic pathological aggregates composed of hyperphosphorylated TDP-43, reminiscent of in vivo TDP-43 pathology. Next, we evaluated the internalization of fluorescent TDP-43 fibrils in wild-type or PrP-overexpressing SH-SY5Y cells, discovering that their uptake was increased when cells overexpressed the prion protein on their membrane. Finally, we tested the effect of TDP-43 fibrils on ScN2a cells observing that, as described for other amyloids, upon treatment with TDP-43 LCD fibrils, these cells presented a reduction in PrP^Sc^ accumulation. Collectively, our results suggest that PrP^C^, which has been already recognized as a common acceptor for Aβ, α-syn and tau oligomeric and amyloid species, presents a similar behavior towards TDP-43 fibrils, expanding the ligand role of PrP^C^ to almost all the major classes of neurodegenerative proteinopathies.

## 2. Materials and Methods

### 2.1. Recombinant TDP-43 LCD Production

The pET-11a plasmid containing human TDP-43 (HuTDP-43(1-414)) coding sequence was purchased from GenScript (Piscataway, NJ, USA). The truncated construct (HuTDP-43(263-414)), or TDP-43 LCD, was obtained by deleting the DNA region encoding for HuTDP-43 fragment 1-262 using mutagenesis. PCR was carried out using site-directed mutagenesis kit (Agilent, Santa Clara, CA, USA) with the following primers: 5′-ttaagaaggagatatacatatgaaacacaacagcaaccgtcagc-3′, 5′-gctgacggttgctgttgtgtttcatatgtatatctccttcttaa-3′. The construct was expressed in *Escherichia coli* BL21 (DE3) cells (Stratagene, La Jolla, CA, USA). Freshly transformed overnight culture was inoculated into Luria Bertani medium with 100 μg/mL ampicillin. At 0.8 OD600, protein expression was induced with isopropyl β-D-1 thiogalactopyranoside (IPTG) to a final concentration of 1 mM. Following overnight incubation at 37 °C, cells were lysed by a homogenizer (PandaPLUS 2000, GEA, Dusseldorf, Germany) and inclusion bodies were washed in a buffer containing 25 mM Tris-HCl, 5 mM ethylenediaminetetraacetic acid (EDTA), 0.8% TritonX100, (pH 8) and then in bi-distilled water (ddH2O) several times. Inclusion bodies were dissolved in five volumes of 8 M guanidine hydrochloride (GdnHCl), loaded onto pre-equilibrated HiLoad 26/60 Superdex 200-pg column, and eluted in 25 mM Tris–HCl (pH 8), 5 mM EDTA, and 5 M GdnHCl at a flow/rate of 1.5 mL/min. Protein refolding was performed by dialysis against 25 mM Tris-HCl (pH 8) using a Spectrapor membrane. Purified protein was analyzed by SDS-PAGE gel electrophoresis under reducing conditions and Western blot.

### 2.2. Brain Homogenate (BH) Preparation

A brain sample (0.5 g) from post-mortem frozen frontal cortex of a patient with a confirmed neuropathological diagnosis of FTLD-TDP associated with *C9orf72* expansion was homogenized as previously described [39]. Briefly, the sample was homogenized in 2.5 mL of homogenization buffer (HB: 10 mM Tris-HCl, 0.8 M NaCl, 1 mM EDTA, 1 mM dithiothreitol (DTT), 1X protease and phosphatase inhibitor cocktail tablets (Roche, Basel Switzerland), pH 7.5). Benzonase (Sigma-Aldrich, St. Louis, MO, USA) was added (0.25 µL) to aliquots of 500 µL, which were then incubated in constant shaking (400 rpm) for 10 min at 37 °C. After this, Sarkosyl was added to each aliquot (final concentration: 1%) and samples were then incubated for 20 min at 37 °C under shaking (400 rpm). Ethanol was added to a final concentration of 20% and samples were incubated in constant shaking (400 rpm) for 10 min at 37 °C. Samples were centrifuged at 150,000× *g* for 60 min at RT. Supernatants were discarded and pellets were suspended in 300 µL PBS 1X by sonication and centrifuged at 150,000× *g* for 60 min at RT. The resulting pellets were suspended in 50 µL of ddH2O by sonication, diluted at 10^−2^ volume/volume in ddH2O and used as seeds for TDP-43 RT-QuIC.

### 2.3. In Vitro Generation of Recombinant Human TDP-43 LCD Aggregates

In vitro aggregation reactions were performed in 200 µL of reaction mix in black, clear-bottom, 96-well microplates. To obtain the unseeded fibrils the reaction mix contained 25 mM Tris at pH 8, 100 mM NaCl, 10 μM ThT, 0.002% of SDS and 0.1 mg/mL of HuTDP-43(263-414). After sealing, the plate was incubated at 40 °C, over a period of 24 h with intermittent cycles of shaking (60 s, 400 rpm, double-orbital) and rest (60 s). To obtain the BH-seeded fibrils the reaction mix contained 25 mM Tris at pH 8, 0.5 M guanidine hydrochloride (GdnHCl) (pH 8), 100 mM NaCl, 10 μM ThT, 0.002% of SDS and 0.05 mg/mL of HuTDP-43(263-414). Reactions were seeded with 20 µL of 10^−2^ diluted BH samples. After sealing, the plate was incubated at 40 °C, over a period of 50 h exposed to 15 s of shaking every 30 min at 100 rpm (double-orbital). All reactions were performed in a FLUOstar OMEGA reader (BMG Labtech, Ortenberg, Germany) and fluorescence intensity, expressed as relative fluorescence units (RFU), was taken every 30 min using 450 ± 10 nm (excitation) and 480 ± 10 nm (emission) wavelengths and a gain of 1000. All solutions were filtered before use with 0.22 µm sterile filters. The addition of a 3 mm glass bead (Sigma) was required to sustain protein aggregation. To avoid toxicity due to the presence of ThT or reagents dissolved in the RT-QuIC reaction buffer, TDP-43 fibrils used for cellular experiments were produced in the absence of ThT and the reaction buffer was removed by means of ultracentrifugation at 186,000× *g* for 1 h. Fibrils were resuspended in sterile PBS 1X and then immersion sonicated for 5 × 5 s bursts with 5 s rests between bursts.

### 2.4. In Vitro Generation of Recombinant Human Tau K18 Aggregates

Recombinant full-length human tau K18 fragment was provided from collaborators in our group. Expression and purification of recombinant tau K18 fragment was performed as previously described [40]. In vitro fibrillization reactions were performed at 37 °C with intermittent cycles of shaking (50 s, 400 rpm, double-orbital) and rest (10 s) in a FLUOstar OMEGA reader (BMG Labtech, Germany). Newly formed aggregates were pelleted by ultracentrifugation (186,000× *g* for 1 h at 4 °C) and resuspended in sterile PBS 1X. Before their use, aliquots were sonicated for 5 min in an ultrasonic bath (Branson 2510, Marshall Scientific, Hampton, VA, USA).

### 2.5. Quantification of TDP-43 LCD and Tau Monomer Incorporated within Fibrils

We measured the starting monomer concentration and its amount left in solution after ultracentrifuging fibrils, using spectrophotometric UV absorbance at 280 nm (Lambda XLS spectrophotometer, Perkin Elmer, Waltham, MA, USA). We then used these values to calculate the per cent monomer incorporated into fibrils and hence, the molar amount of protein in fibrils: (([initial monomer] − [monomer at end stage])/[initial monomer]) × 100. We found that ~90% of starting TDP-43 and tau K18 were incorporated into fibrils. Using this information, we estimated their molar concentrations.

### 2.6. Transmission Electron Microscopy Analyses

Ten μL of RT-QuIC products were dropped onto 200 mesh Formvar-carbon coated nickel grids for 30 min and the remaining drop was blotted dry using filter papers. The grids were subsequently stained with 25% Uranyl Acetate Replacement (UAR, Electron Microscopy Sciences, Hatfield, IN, USA) for 15 min, the solution was removed using filter papers and the grids were air-dried for 15 min before the analyses. Images were recorded at 120 kV with a FEI Tecnai Spirit, equipped with an Olimpus Megaview G2 camera.

### 2.7. In Vitro Assay for PrP-TDP-43 LCD Fibril Interaction

Before their use, full-length recombinant (rec) MoPrP and monomeric TDP-43 LCD were ultracentrifuged at 186,000× *g* for 1 h at RT to remove precipitates and only the supernatants were used to perform the assay. One μg of full-length mouse recPrP (~0.01 μM) was diluted in a final volume of 500 μL PBS 1X and exposed at a 1:50 molar ratio to either recombinant monomeric TDP-43 LCD or to sonicated TDP-43 LCD unseeded fibrils. As a control we used recombinant monomeric tau K18 and tau K18 end-stage sonicated fibrils at the same concentrations. After 30 min of incubation at RT we separated and collected the soluble and pellet fractions of the reaction by ultracentrifugation at 186,000× *g* RT. These fractions were subsequently analyzed by Western blotting (WB) using the anti-PrP antibody W226 [41].

### 2.8. SH-SY5Y and N2a Cell Line Generation

The pcDNA3.1 plasmid encoding full-length human and mouse PrP were purchased by GenScript. All transfection reactions were performed using Effectene Transfection Reagent (Qiagen, Milan, Italy) according to the manufacturer guidelines. Cells were plated in a volume of 10 mL of F12-DMEM complete medium in a 10 cm^2^ plate 24 h before transfection. The transfection was performed when cells reached 30–40% confluency. On the day of transfection, 0.4 μg of DNA were diluted in Buffer EC, to a total volume of 100 μL. After this, 3.2 μL Enhancer were added and the solution was mixed by vortexing for 1 s. Following a 5 min incubation at RT, 10 μL Effectene Transfection Reagent were added to the DNA-Enhancer mixture and incubated for a further 10 min at RT. DNA transfection mixture was added to the cells in a dropwise manner. Cells were incubated with DNA transfection mixture for 18 h at 37 °C, following which, fresh medium was added in replacement. Two days after transfection cells were split into fresh medium containing 1mg/mL Geneticin (Thermo Fisher Scientific, Whaltham, MA, USA). The selective medium was changed every 3–4 days until Geneticin-resistant foci could be identified. To perform foci collection, almost all the medium from the plate was removed, leaving only 1–2 mL. After that, single clones were isolated using sterile plastic tubes. Once a clone was isolated within the tube, the remaining medium was removed and Trypsin (50 μL) was added. After incubation for 1–2 min at 37 °C 100 μL of medium were added and pipetted to resuspend detached cells. Collected cells from a single clone were plated in a new 10 cm^2^ plate with the addition of 1mg/mL Geneticin. After that, stable cell lines were maintained in medium containing 400 µg/mL Geneticin. The PrP- overexpressing N2a cell line that we used had been already generated in our lab using the same protocol described above. N2a PrP^-/-^ cells were kindly provided by professor Gerold Schmitt-Ulms (Tanz Centre for Research in Neurodegenerative Diseases, University of Toronto, Toronto, Ontario, Canada), for which they used the CRISPR-Cas9-Based Knockout system to ablate the expression of PrP protein [42].

### 2.9. Cell Vitality Assay (MTT)

Cells were seeded to a concentration of 1 × 10^4^ in a 96-well, tissue culture-treated, clear bottom, plate. Cells were allowed to grow for 24 h at 37 °C under 5% CO_2_ prior to the addition of unseeded or BH-seeded TDP-43 LCD fibrils (prepared as described above), monomeric recombinant TDP-43 LCD or equivalent amounts of sterile PBS 1X, as a control. TDP-43 fibrils were diluted in the cell medium to a final concentration of 1 µM for the unseeded fibrils and monomeric TDP-43 LCD or 500 nM for BH-seeded fibrils. After treatment, cells were cultured at 37 °C under 5% CO_2_ for 24 h. 3-(4,5-dimethylthiazol- 2-yl)-2,5-diphenyltetrazolium bromide (MTT, Sigma-Aldrich) was diluted in PBS 1X to a working solution of 5 mg/mL. Cells were incubated with the MTT solution for 3 h at 37 °C under 5% CO_2_. After incubation, cell medium was removed and a solution of DMSO-2-Propanol (1:1) was added to each well and the plate was kept at RT for 5 min before reading. The absorbances of the samples was measured using EnSight Multimode plate reader (Perkin Elmer) at 570 nm and the reference wavelength is 690 nm. Each condition was tested in six wells and in at least four independent experiments.

### 2.10. TDP-43 Fibril Conjugation with AlexaFluor-488

TDP-43 fibrils were conjugated to AlexaFluor-488 NHS Ester (Thermo Fisher Scientific) according to the manufacturer’s instructions. Briefly, TDP-43 fibrils were resuspended, in 0.1 M sodium bicarbonate buffer, pH 8.3 at a concentration of 2 mg/mL, after immersion sonication (5 pulses of 5 s each). After the addition of 50 µg of the reactive dye, the reaction mix was incubated for 1 h at RT with continuous stirring. The unbound dye was removed with 3 subsequent dialysis against sterile PBS 1X and the conjugated fibrils were aliquoted and stored at −20 °C.

### 2.11. Treatments of SH-SY5Y Cells with TDP-43 LCD Fibrils

TDP-43 LCD fibrils were added to WT or PrP-overexpressing SH-SY5Y cell culture medium at a final concentration of 1 µM (unseeded, AlexaFluor-488-conjugated) or of 1 µM or 500 nM with or without lipofectamine according to the type of experiment (BH-seeded). Cells were processed for immunofluorescence and analyzed at the confocal microscopy at different timepoints (24 or 72 h) and incubation times (6, 24 or 72 h), according to the type of experiments, as specified in the main text and in each figure legend. For immunofluorescence analysis performed after 6 h of incubation and 24 h as final timepoint, before fixing with paraformaldehyde, cells were incubated 5 min with trypan blue (diluted 1:1 in PBS 1X) to quench fibrils present outside cell bodies. Trypan Blue quenches green fluorescence through an energy transfer mechanism, but being unable to enter alive cells, only the fluorescence coming from non-internalized fibrils is quenched [30]. For all experiments, cells were fixed in 2% paraformaldehyde (40 min) at 37 °C, washed with PBS 1X and permeabilized for 5 min with 0.1% Triton X-100 prior to 30 min blocking in 7% normal goat serum (NGS), 5% bovine serum albumin (BSA), 0.1% Triton X-100. Cells were incubated for 2 h at 4 °C with primary antibodies and after three washes with PBS 1X incubated with the respective secondary antibodies conjugated with Alexa Fluor. After three washes with PBS 1X cells were incubated 5 min with HCS Blue CellMask (Thermo Fisher Scientific) (1:5000 dilution), a specific dye that labels the whole-cell cytoplasm or with DAPI (1:1000 dilution), depending on the experimental settings. Coverslips were mounted in Fluoromount-GTM (Thermo Fisher Scientific) and stored at 4 °C for confocal fluorescence microscopy. Images were acquired using a Nikon confocal microscope (Nikon A1R). Primary antibodies: rabbit polyclonal anti-TDP-43 C-terminal (Sigma-Aldrich-T1580), rat monoclonal anti-phospho TDP-43 (Ser409/Ser410), clone 1D3 (Sigma-Aldrich-MABN14).

### 2.12. Treatment of ScN2a Cells with TDP-43 Fibrils

8.5 × 10^4^ ScN2a cells were seeded in a 6 cm plate. The day after, the medium was replaced and monomeric TDP-43 LCD (2 µM), TDP-43 unseeded (2 µM) or BH-seeded (1 µM) fibrils were added. Cells were kept in culture for 72 h after which, were rinsed with PBS 1X and resuspended in lysis buffer (10mM Tris HCl, 150mM NaCl, 0.5% NP-40, 0.5% Sodium deoxycholate). The total protein amount was assayed using Quantum protein assay kit (Euroclone, Milan, Italy). To assess the amount of PrP^Sc^ in fibrils-treated samples, cell lysates were subjected to proteinase K digestion as follows: 150 µg of total proteins were incubated for 1 h at 37 °C with 20 µg/mL of proteinase K. The reaction was stopped adding 2 mM of phenylmethylsulphonyl fluoride (PMSF, Sigma-Aldrich) followed by a centrifugation at 186,000× *g* for 1 h at 4 °C. The resulting pellet was resuspended in 20 µL of Loading Buffer (10% Glycerol, 50 mM Tris-HCl, 2% Sodium dodecyl sulfate, 4M Urea, Bromophenol blue and fresh-added 200 mM Dithiothreitol) and boiled for 10 min before SDS PAGE.

### 2.13. Western Blot

Samples were loaded onto 12% Tris-Glycine SDS-PAGE and transferred onto Immobilon P PVDF membranes (Millipore, Burlington, MA, USA) for 2 h at 4 °C. Membranes were blocked in 5% non-fat milk in TBST (Tris 200 mM, NaCl 1.5 mM, 1% Tween-20, Sigma-Aldrich) for 1 h at RT. The following primary antibodies were incubated overnight at 4 °C: anti-PrP W226 1:1000 [41] and anti-PrP EB8 1:1000 [43]. After three washes in TBST, membranes were incubated with goat anti-mouse IgG conjugated with horse-radish peroxidase for 1 h at RT and developed using Immobilon Classico Western HRP substrate (Millipore). Anti-β-actin antibody 1:10,000 (Sigma-Aldrich) was incubated for 1 h at RT and used for normalization. Images were acquired with Uvitec Alliance (Cambridge, UK) and Uviband analysis software was used for densitometry analysis.

### 2.14. Fluorescence-Activated Cell Sorting (FACS)

Before flowcytometry analysis, cells were trypsinized and resuspended in FACS buffer (1% FBS; 25 mM HEPES in PBS 1X). Cell suspension was passed through a 40-mm strainer (Biorad, Hercules, CA, USA) to form single-cell suspension and then cells were loaded to S3e cell sorter (Biorad).

### 2.15. PCR and RT-qPCR Analysis

Total RNA from each sample was purified using TRIzol reagent (Thermo Fischer Scientific, Waltham, MA, USA) according to manufacturer instructions. The cDNA synthesis was performed using 1 μg of each RNA with 50 μM Oligo(dT)20, 10 mM dNTP mix, 5X First Strand Buffer, 0.1 M DTT, 40 U RNAse inhibitor, and 200 U SuperScript III Reverse Transcriptase (Thermo Fischer Scientific). PCR analysis was performed using the following primers: *POLDIP3* FW 5′-gcttaatgccagaccgggagttg-3′; *POLDIP3* RV 5′-tcatcttcatccaggtcatataaatt-3′; *STAG2* FW 5′-gtatgtttacttggaaaagttcatg-3′; *STAG2* RV 5′-tgattcatccataattgaagctgga-3′. Specific qPCR primers were designed using the online tool Primer-Blast provided by NCBI. The primer sequences were as follows: for *GAPDH* FW 5′-cctgcaccaccaactgctta-3′ and RV 5′-tcttctgggtggcagtggatg-3′; *TARDBP* FW 5′-cagcttcggagagttctggg-3′; *TARDBP* RV 5′-cagcaaaccgcttgggatta-3′. Gene expression assays were performed using iQ^TM^ SYBR Green Supermix 2x (BioRad), 400nM final concentration of the corresponding forward and reverse primers (Sigma-Aldrich) and 10 ng/μL final concentration of cDNA samples, with CFX96 Touch^TM^ Real-Time PCR Detection System (BioRad).

### 2.16. Statistical Analysis

Statistical analyses were performed using GraphPad Prism software (version 8.4.3; San Diego, CA, USA). Values were expressed as mean ± standard deviation (SD). Since for all our analyses the “n” was too small, we applied only non-parametric tests. Each performed test and the corresponding *p* value is reported in the corresponding figure legend throughout the paper.

## 3. Results

### 3.1. TDP-43 LCD Fibrils Bind to PrP^C^

We recently optimized and described a TDP-43 RT-QuIC which allows the aggregation of monomeric recombinant TDP-43 into fibrils under controlled and standardized conditions [39]. To increase inter-laboratory reproducibility of our results, we decided, as others [21], to avoid using intermediates of fibrillogenesis and to collect end-stage TDP-43 fibrils to perform all our analyses. Since the LCD of TDP-43 (amino acids 263–414) is the aggregation-prone prion-like domain of the protein and is considered to be responsible for TDP-43 protein-protein interactions [4], we focused on this fragment and we collected and characterized the final product of its fibrillization under spontaneous aggregation (unseeded fibrils) (Figure 1a). We applied the same calculation proposed by Corbett and colleagues [21] to estimate the percentage of the monomeric protein internalized within end stage fibrils (see materials and methods section), which was ~90%. When we tried, after sonication, to separate a soluble and insoluble fibril fraction, we realized that almost all TDP-43 LCD sonicated fibrils contained in the mixture precipitated again in the pellet after centrifugation, suggesting that a soluble fraction was not represented in our sample. For this reason, all our study was conducted only with the total fraction of TDP-43 LCD sonicated fibrils. To evaluate the binding between TDP-43 fibrils and recombinant PrP^C^ (recPrP) we devised a specific in vitro assay (Figure 1b). Full-length mouse recMoPrP (~0.01 μM) was exposed at a 1:50 molar ratio to either recombinant monomeric TDP-43 LCD or sonicated TDP-43 LCD unseeded fibrils (both at ~0.5 μM concentration). As a control we used recombinant monomeric tau K18 and tau K18 end-stage sonicated fibrils at the same concentrations. After 30 min of incubation, we separated and collected the soluble and pellet fractions of the reaction by ultracentrifugation. These fractions were subsequently analyzed by Western blotting (WB) using the anti-PrP antibody W226. Our hypothesis was that if recMoPrP and TDP-43 (or tau K18) fibrils interacted, the excess of insoluble TDP-43 LCD (or tau K18) fibrils bound to monomeric and soluble recMoPrP would pull it down in the pellet fraction upon ultracentrifugation. On the contrary, recMoPrP exposed to soluble monomeric TDP-43 LCD (or tau K18), after ultracentrifugation would remain in the soluble fraction, regardless of its interaction with monomeric and soluble TDP-43 (or tau K18) (Figure 1b). The amount of PrP^C^ detected in the pellet fraction of the reaction performed with TDP-43 LCD fibrils was significantly higher than the one carried out with the monomeric TDP-43 counterpart (Figure 1c). We observed the opposite results in the soluble fractions, in which PrP^C^ content was reduced in the soluble fraction exposed to TDP-43 fibrils (Figure 1d). The results of pellet and soluble fractions collected after exposure to monomeric or fibrillary tau K18 showed the same trend (Figure 1c,d). These results suggest that TDP-43 LCD fibrils interact with the recombinant full-length prion protein in solution.

### 3.2. PrP^C^ on the Cell Surface Increases TDP-43 LCD Fibrils Toxicity in Both SH-SY5Y and N2a Cells

The expression levels of endogenous PrP^C^ in SH-SY5Y cells are extremely low (Appendix A), and so wild-type (WT) SH-SY5Y have been already exploited in previous studies as a natural PrP^C^ knock-down cell line [17,18]. Moreover, WT SH-SY5Y cells express detectable levels of the *TARDBP* gene, encoding for TDP-43 (Appendix A), a key requirement to observe a fibril-induced intracellular seeding reaction. We stably transfected SH-SY5Y cells to express high levels of the full-length mouse (Mo) PrP^C^ (designated as over-MoPrP) and we selected the most overexpressing clone to proceed with our analysis (Appendix A).

Treatment with 1 μM TDP-43 LCD unseeded fibrils induced a reduction in cell vitality (expressed as the percentage of cells with an active metabolism) in over-MoPrP, while it presented only a mild toxic effect on WT SH-SY5Y cells (Figure 2a). We observed the same results when we applied this treatment to a different cell line, N2a cells (Figure 2b). Since WT N2a cells express high levels of PrP^C^, in this case we used PrP-knock-out (PrP^-/-^) and MoPrP-overexpressing cells (over-MoPrP) (Appendix A). As shown in Figure 2b, also in this case N2a PrP^-/-^ cells were not susceptible to treatment with 1 μM TDP-43 LCD unseeded fibrils, while cell vitality was impaired in over-MoPrP N2a cells. Next, we tested the toxic effect of TDP-43 LCD fibrils seeded with an autopsy-verified patient-derived brain homogenate sample (BH-seeded) (Appendix A) and we observed again a significant reduction in the vitality of over-MoPrP SH-SY5Y and N2a cells compared respectively to WT-SHSY5Y and PrP^-/-^ N2a cells (Figure 2c,d). For both cell lines (SH-SY5Y and N2a cells), BH-seeded fibrils showed higher degrees of toxicity compared to the unseeded ones, even if they were used at lower concentrations (500 nM BH vs. 1 μM unseeded) (Figure 2a–d). Cell vitality presented a more significant reduction in over-MoPrP SH-SY5Y cells (67.92 ± 7.601%, unseeded; 56.99 ± 3.210%, BH-seeded; mean ± standard deviation, SD) than in N2a cells overexpressing PrP (70.05 ± 6.993%, unseeded; 63.58 ± 8.995%, BH seeded; mean ± SD). Interestingly, among SH-SY5Y cells, the difference between the toxic effect of BH-seeded and unseeded fibrils was enhanced in over-MoPrP cells (Figure 2e,f). When we treated SH-SY5Y cells with the monomeric form of TDP-43 LCD (1 μM) we observed no signs of toxicity in both WT and over-MoPrP cell lines (Figure 2g). Transmission electron microscopy (TEM) analysis showed that unseeded and BH-seeded fibrils differed in terms of structural organization. While the BH-seeded fibrils acquired a typical fibrillary structure the unseeded one presented a different organization characterized by bundles of fibers associated with the presence of less fibrillar-structured material (Figure 2h and Appendix A). Taken together these results indicate that the presence of PrP^C^ on the cell surface of both SH-SY5Y and N2a cells increases the toxic effect of TDP-43 LCD fibrils. Furthermore, it is possible that the specific morphology of BH-seeded fibrils is responsible for their increased toxicity since their conformation may favor their interaction with PrP^C^ on the cell surface.

### 3.3. Intracellular Uptake and Phosphorylation of TDP-43 LCD Fibrils in WT and PrP-Overexpressing SH-SY5Y Cells

As a first attempt to observe the internalization of TDP-43 aggregates within WT-SH-SY5Y cells, TDP-43 LCD BH-seeded fibrils were sonicated, mixed with lipofectamine and added to the cell culture medium to a final concentration of 500 nM. At one and three days after treatment we observed the presence of TDP-43 aggregates within the cytoplasm of exposed cells (Figure 3a, white arrowheads). Notably, some of the aggregates stained using the C-terminal anti-TDP-43 antibody were also positive for the phospho-specific anti-TDP-43 one (Figure 3a, yellow arrowheads). The observation of phosphorylated TDP-43 aggregates suggested that in vitro obtained fibrils (intrinsically devoid of post-translational modifications due to bacterial expression of the monomeric recombinant protein used as substrate for the RT-QuIC) were internalized in cells and actively phosphorylated, a feature reminiscent of in vivo TDP-43 pathology. This observation is also in keeping with previous data showing that C-terminal fragments of TDP-43 become heavily phosphorylated when expressed in cells [44].

We evaluated the toxic effect of this treatment observing that it resulted in a severe reduction of cell vitality (~50%) (Figure 3b). One of the hypothesized mechanisms of toxicity of TDP-43 aggregates within neurons is that TDP-43 accumulation in the cytosol is accompanied by its loss of function in the nuclear compartment, with downstream alterations of several genes under its control [4]. To investigate if this was the case in our cellular model, we evaluated TDP-43 function on two splicing targets [45], namely *STAG2* and *POLDIP3* transcripts and we observed no differences in the representation of their splicing variants between treated and control cells (Figure 3c), suggesting that TDP-43 function was maintained in treated cells.

Next, to observe the spontaneous internalization of TDP-43 fibrils, we repeated the previous experiment in a more physiological condition, i.e., without the addition of lipofectamine. TDP-43 LCD unseeded fibrils were sonicated and conjugated with the AlexaFluor-488 green-fluorescent dye and added to both WT and over-MoPrP SH-SY5Y cell culture medium to a final concentration of 1 μM. After 6 h of incubation the cell medium was changed, and cells were kept in culture for 72 h and then analyzed by means of immunofluorescence. Cells were stained with the cytoplasmic marker CellMask which produces a clear delimitation of cellular bodies, facilitating our evaluation of fibril internalization (Figure 3d,e). Even in the absence of lipofectamine, we observed that TDP-43 fibrils were abundantly internalized in both WT and over-MoPrP SH-SY5Y cells and that some of the intracellular aggregates were positively stained by the phospho-specific anti-TDP-43 antibody (Figure 3f,g). We observed the same results when we treated cells with the same concentration of BH-seeded TDP-43 LCD fibrils (Appendix A).

### 3.4. PrP^C^ Boosts TDP-43 LCD Fibril Internalization in over-PrP-SH-SY5Y Cells

To test if the presence of PrP^C^ on the cell surface modifies TDP-43 fibril uptake we used cell sorting technology to quantify the amount of WT and over-MoPrP SH-SY5Y cells harboring internalized fluorescent fibrils after treatment (Figure 4a). WT and over-MoPrP SH-SY5Y cells were exposed to either 1 μM of AlexaFluor-488 green fluorescent TDP-43 LCD unseeded fibrils or PBS 1X. After 6 h, cell culture medium was changed, cells were kept in culture for 24 h and then trypsin-detached, resuspended in PBS 1X and analyzed with the cell sorter. Cells exposed to PBS 1X were used as control for the cell sorter to set the threshold of background fluorescence. Among cells treated with fluorescent fibrils, we observed a significantly higher proportion of cells containing internalized fluorescent fibrils in the over-MoPrP population (~5% of the total) compared to WT cells (~1.5% of the total) (Figure 4a). Results of this analysis strongly indicate that PrP^C^ present on the cell surface facilitates the uptake of TDP-43 fibrils. Since we observed that internalized fibrils induced the formation of intracytoplasmic deposits of hyperphosphorylated TDP-43 (Figure 3), we hypothesize that the increased toxicity of TDP-43 fibrils that we observed in PrP^C^-overexpressing cells, was linked to their enhanced PrP-mediated internalization.

### 3.5. Validation of PrP^C^ Role on SH-SY5Y Cells Overexpressing the Human Form of the Prion Protein

Despite mouse and human PrP^C^ sequences sharing a high degree of homology [46], their tertiary structures do not completely overlap, so their expression in a human cell line could result in different biological downstream effects.

For this reason, we decided to replicate our main findings on SH-SY5Y cells overexpressing the human (Hu) form of PrP^C^. With this aim, we generated SH-SY5Y cells overexpressing HuPrP^C^ (over-HuPrP) and selected the most overexpressing clone (Figure 4b). Immunofluorescence analysis confirmed that over-HuPrP cells internalized AlexaFluor-488 green fluorescent TDP-43 LCD unseeded fibrils and displayed, as WT and over-MoPrP SH-SY5Y cells, the presence of intracellular aggregates composed of phosphorylated TDP-43 (Figure 4c). Next, we treated over-HuPrP SH-SY5Y cells with 1 μM of AlexaFluor-488 green fluorescent TDP-43 LCD unseeded fibrils and measured their uptake at the cell sorter. Our results unambiguously confirmed that the presence of high levels of HuPrP^C^ on the cell surface increases the uptake of TDP-43 fibrils (Figure 4d). When we checked for TDP-43 fibril toxicity we observed a different scenario compared to the one resulting from treatment of over-MoPrP SH-SY5Y cells. The fibril conformational toxic effect was maintained also after treatment of over-HuPrP SH-SY5Y cells (500 nM BH seeded fibrils were more toxic than 1 μM of the unseeded ones; Figure 4h) but the overall fibril toxicity was reduced (Figure 4e,f,h–j). Also in this case, treatment with the monomeric form of TDP-43 LCD did not affect cell vitality (Figure 4g). Taken together our data suggest that the presence of HuPrP^C^ on the cell surface of SH-SY5Y cells led to multiple consequences: on one hand it promoted the internalization of TDP-43 fibrils, on the other it might have exerted a protective and trophic effect towards cell vitality, resulting ultimately in the paradox of increased fibril uptake associated with lower signs of toxicity (Figure 4d–j). One possible explanation to the apparent controversial nature of the different TDP-43 fibril toxicity observed between over-HuPrP and over-MoPrP SH-SY5Y cells (Figure 4i,j) could be that mouse PrP^C^ expressed on the cell surface of a human cell line might not display the necessary structural equipment to approach and contact all the specific PrP-binding partners [47,48] required to exert its predicted protective role(s) [49].

### 3.6. TDP-43 Fibrils Reduce PrP^Sc^ Accumulation in Prion Infected Cells

Our group, and others, previously described a peculiar interplay among Aβ, α-syn and tau fibrils and PrP^Sc^ accumulation, resulting in decreased PrP^Sc^ levels in cells chronically infected with prions upon treatment with these amyloids [30,34,38]. One hypothesis to explain these results is that if these misfolded species were able to bind to PrP^C^ on the cell surface, they could act as competitive inhibitors, reducing PrP^C^-PrP^Sc^ interaction, thus resulting in a decreased conversion rate of PrP^C^ into its pathological counterpart, PrP^Sc^. To verify if this reduction in PrP^Sc^ levels could be induced also by TDP-43 fibrils, we exposed RML chronically infected N2a cells (ScN2a RML) to either 1 μM BH-seeded TDP-43 LCD fibrils or PBS 1X and analyzed the PK-resistant PrP^Sc^ levels at 72 h. The quantification of four independent experiments showed that upon exposure to BH-seeded TDP-43 LCD fibrils the accumulation of PrP^Sc^ presented ~30% of reduction (Figure 5a).

Interestingly, to observe comparable results using TDP-43 LCD unseeded fibrils we increased their concentration two-folds (Figure 5b). These results are in line with the observed toxic effect of these two distinct fibril species and suggest that BH-seeded fibrils probably display higher PrP-binding affinity. Unexpectedly, when we performed the same experiment using the TDP-43 LCD monomeric form (2 μM) as a control treatment, we observed a reduction in PrP^Sc^ levels comparable to the one induced by TDP-43 LCD BH-seeded fibrils (Figure 5c). We observed that only after treatment with monomeric TDP-43 the reduction in PrP^Sc^ levels was accompanied by an almost equal diminution of the levels of the total PrP fraction (calculated as the total amount of PrP in cells before digestion with proteinase K) (Figure 5d). A detailed characterization of this phenomenon goes beyond the scope of our current study. However, one possible interpretation of these results is that the observed PrP^Sc^ clearance induced by treatment with monomeric TDP-43 was not related to the interaction between PrP^C^ and monomeric TDP-43 on the cell surface but was instead connected to a reduction in PrP^C^ expression levels in the cell.

## 4. Discussion

Increasing evidence supports the idea that the cell surface prion protein can act as a common acceptor for the pathological misfolded forms of multiple proteins involved in neurodegeneration [17,18,19,20,21,22,23,24,25,26,27,28,29,30,31,32,33,34,35,36,37] but so far the interaction between PrP^C^ and TDP-43 fibrils has not been investigated.

Here, capitalizing on our newly adapted TDP-43 RT-QuIC [39], we generated and characterized two TDP-43 fibril species obtained by either the self-aggregation of the TDP-43 LCD substrate (unseeded) or by its fibrillization in the presence of a patient-derived BH sample containing misfolded TDP-43 (BH-seeded). We show that these in vitro produced TDP-43 LCD fibrils interact with recombinant PrP^C^ and are internalized in WT SH-SY5Y cells, leading to the formation of intracytoplasmic aggregates composed of hyperphosphorylated TDP-43. Although this phenomenon occurred also in WT SH-SY5Y cells, we proved that TDP-43 fibril uptake is significantly increased when cells express high levels of PrP^C^ on their surface. SH-SY5Y cells showed higher TDP-43 fibril internalization when overexpressing either the mouse or human form of the prion protein. We observed that this increased internalization of TDP-43 fibrils resulted in a reduced cell vitality in over-MoPrP compared to WT SH-SY5Y cells. These results were verified also on a different cell model, represented by N2a cells knocked-out for the prion protein or engineered to express it at high levels. BH-seeded TDP-43 species presented the highest degree of toxicity in all our cell lines. Furthermore, TEM analysis showed that they presented a different organization respect to the unseeded ones. These data confirm that, as other groups reported [14,15], misfolded TDP-43 can also acquire different pathological conformations (i.e., strains) associated with specific biological features, as other prion-like proteins. Since, in our experimental setting, it was not possible to separate a soluble fraction from the total fibril preparation our data are not informative on which specific subtype of TDP-43 fibrils (i.e., insoluble, soluble/oligomeric) is preferentially recognized by PrP^C^.

We realized that overexpressing HuPrP^C^ on the cell surface of SH-SY5Y cells resulted in a more complex scenario than the one observed after MoPrP^C^ expression. It increased the uptake of TDP-43 fibrils, but it also presented a sort of protective effect towards fibril toxicity. We hypothesize that the human form of PrP^C^ presents the proper structural equipment, which could be absent in MoPrP^C^, allowing the specific interactions with its binding partners [47,48] required to express its known protective effects [49]. In this regard, it should be stressed that we selected SH-SY5Y cells as the best candidate cell line to perform our set of experiments because of several specific features, among which the most important one was represented by their natural low levels of expression of PrP^C^ on the cell surface. This was the best strategy to explore PrP^C^ role on TDP-43 fibril uptake, but probably does not still represent the most adequate model to evaluate its involvement in its whole complexity. Nonetheless, the main goal of our study was indeed to perform a preliminary evaluation on PrP^C^ and TDP-43 fibril interaction and the most striking result that we obtained was that, as for other amyloids, TDP-43 fibril internalization is also facilitated by PrP^C^. We are aware that to further investigate PrP^C^ role(s) in the transmission of TDP-43 fibril toxicity other more complex models should be exploited.

Finally, we observed that treatment with TDP-43 fibrils induced a reduction in PrP^Sc^ accumulation in chronically prion infected N2a cells. This result is in line with previous studies performed by our and other groups [30,34,38]. Treatment with the monomeric form of TDP-43 LCD induced a similar decrease of PrP^Sc^ levels in ScN2a cells. In the attempt to explain this unexpected result, we evaluated the levels of total PrP in ScN2a treated cells and observed that only upon treatment with monomeric TDP-43 LCD, PrP^Sc^ clearance was paralleled by an equal reduction in total PrP levels. This suggests that the mechanism by which monomeric TDP-43 LCD induced the observed PrP^Sc^ reduction could be a specific one, different from the one hypothesized for TDP-43 fibrils. A detailed characterization of the mechanism(s) behind this phenomenon goes beyond the scope of our current study, but we speculate that a likely scenario involves a TDP-43-mediated reduction in PrP levels of expression. This hypothesis is also supported by the fact that TDP-43 is an RNA-binding protein involved in many regulatory processes connected to RNA splicing, maturation and translation [50]. Therefore, although we administered only the C-terminal fragment of TDP-43 devoid of its RNA recognition motifs, it is not unlikely that the LCD, responsible for many protein-protein interactions, could have mediated unexpected downstream biological effects.

In conclusion, we have investigated for the first time the possible link between PrP^C^ expression on the cell surface and the internalization of TDP-43 LCD fibrils and have observed that their uptake was increased in cells overexpressing both human and mouse prion protein. This increased internalization was associated with detrimental consequences in all PrP-overexpressing cell lines but was milder in SH-SY5Y cells overexpressing the human form of the prion protein. Besides confirming literature data regarding PrP^C^ beneficial and protective functions, our study expands the list of misfolded proteins whose uptake and detrimental effects are PrP^C^-mediated to encompass almost all the pathological amyloids involved in neurodegeneration. Our results support the idea of a complex and multifaceted spectrum of PrP^C^ functions and suggest that therapeutic strategies aimed at targeting PrP^C^ should consider the complexity of this molecule in its whole to avoid undesired collateral effects.

## Figures and Tables

**Figure 1 viruses-13-01625-f001:**
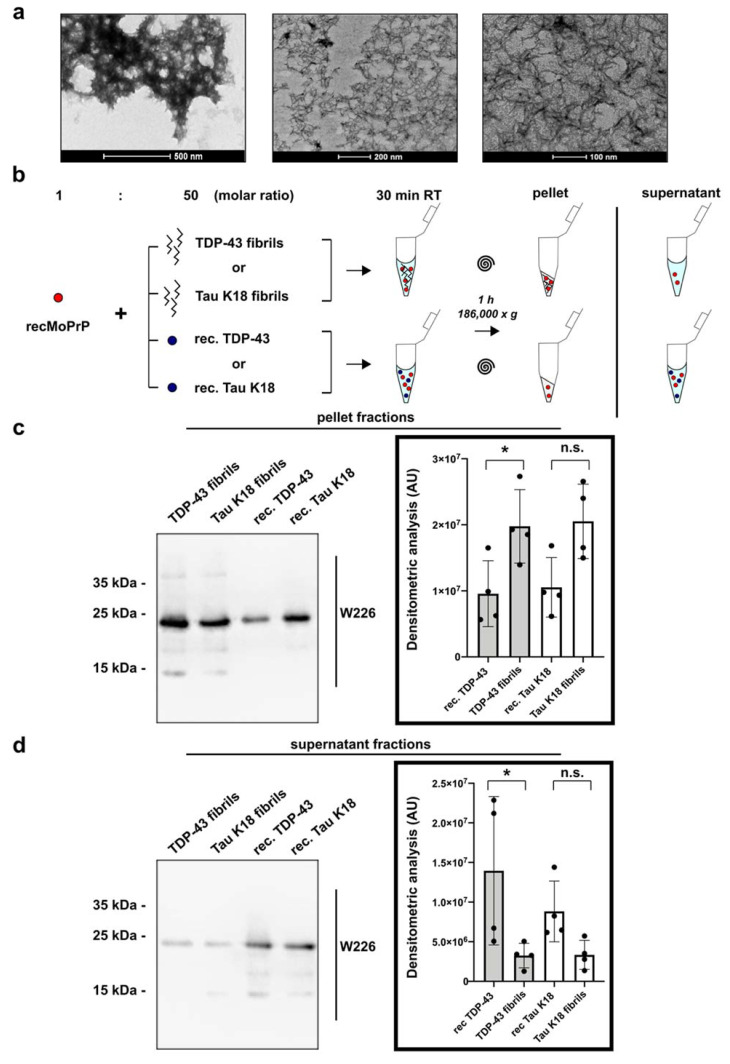
In vitro interaction between PrP^C^ and TDP-43 LCD unseeded fibrils. (**a**) Transmission electron microscopy analysis of TDP-43 LCD unseeded fibrils collected when the reaction reached its plateau; (**b**) Schematic representation of the possible readouts for the in vitro assay designed to explore the binding between recombinant full-length mouse PrP^C^ (recMoPrP) and TDP-43 LCD fibrils; (**c**,**d**) representative WB of the pellet and supernatant fractions collected after ultracentrifugation and analyzed with the anti-PrP W226 antibody; the right part of the image shows the graphs with the corresponding quantifications. Data were evaluated with Friedman test and Dunn’s multiple comparisons; * *p* < 0.05; n.s.: not significant. Data in the graphs are reported as mean ± SD; each dot represents an independent experiment.

**Figure 2 viruses-13-01625-f002:**
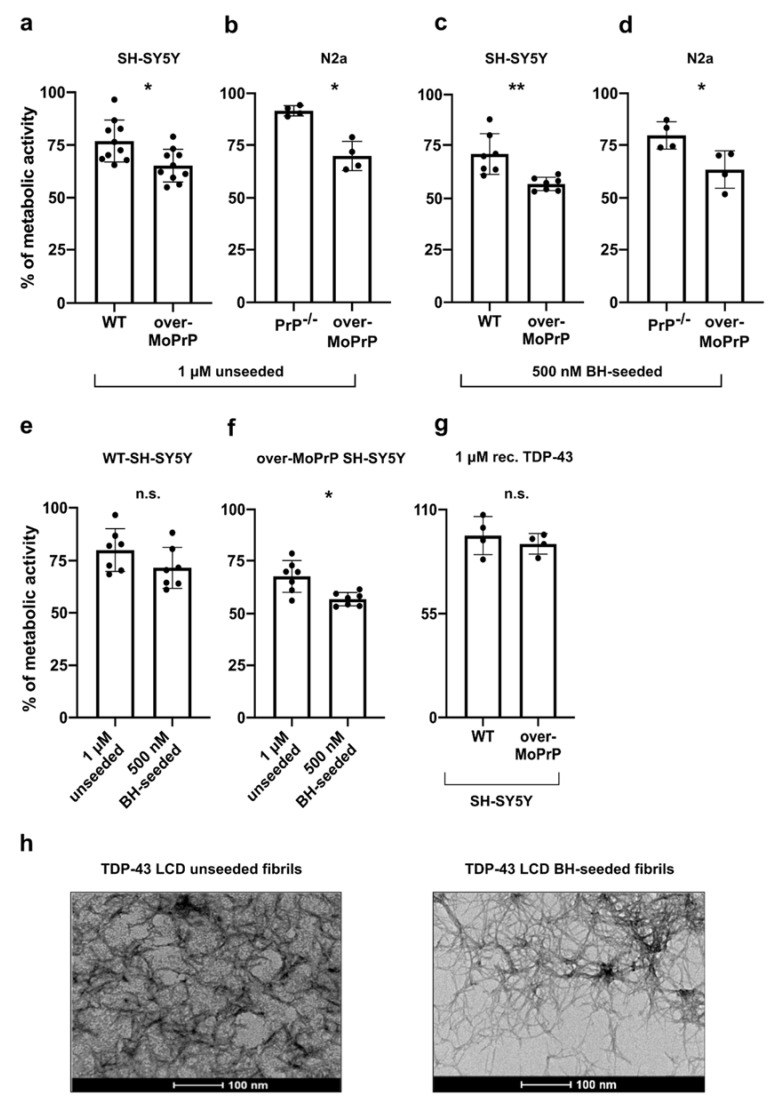
PrP^C^ on the cell surface increases TDP-43 LCD fibril toxicity in SH-SY5Y and N2a cells. Cell vitality of WT or over-MoPrP SH-SY5Y and PrP^-/-^ or over-MoPrP N2a cells. after treatment with (**a**,**b**) 1 μM unseeded or (**c**,**d**) 500 nM BH-seeded TDP-43 LCD sonicated fibrils; comparison of the toxic effect of unseeded and BH-seeded TDP-43 LCD fibrils on WT (**e**) and over-MoPrP (**f**) cells; (**g**) cell vitality of WT and over-MoPrP SH-SY5Y cells after treatment with 1 μM recombinant monomeric TDP-43 LCD; data were evaluated with Mann-Whitney test; * *p* < 0.05; ** *p* < 0.01; n.s.: not significant. Cell vitality was evaluated as the percentage of treated cells with an active metabolism at the MTT assay compared to control cells treated with 1X PBS (corresponding to the 100% value of metabolic activity). Data in the graphs are reported as mean ± SD; each dot represents the mean value for the six replicates of each independent experiment; (**h**) transmission electron microscopy analysis of TDP-43 LCD unseeded and BH-seeded fibrils collected when the reaction reached its plateau.

**Figure 3 viruses-13-01625-f003:**
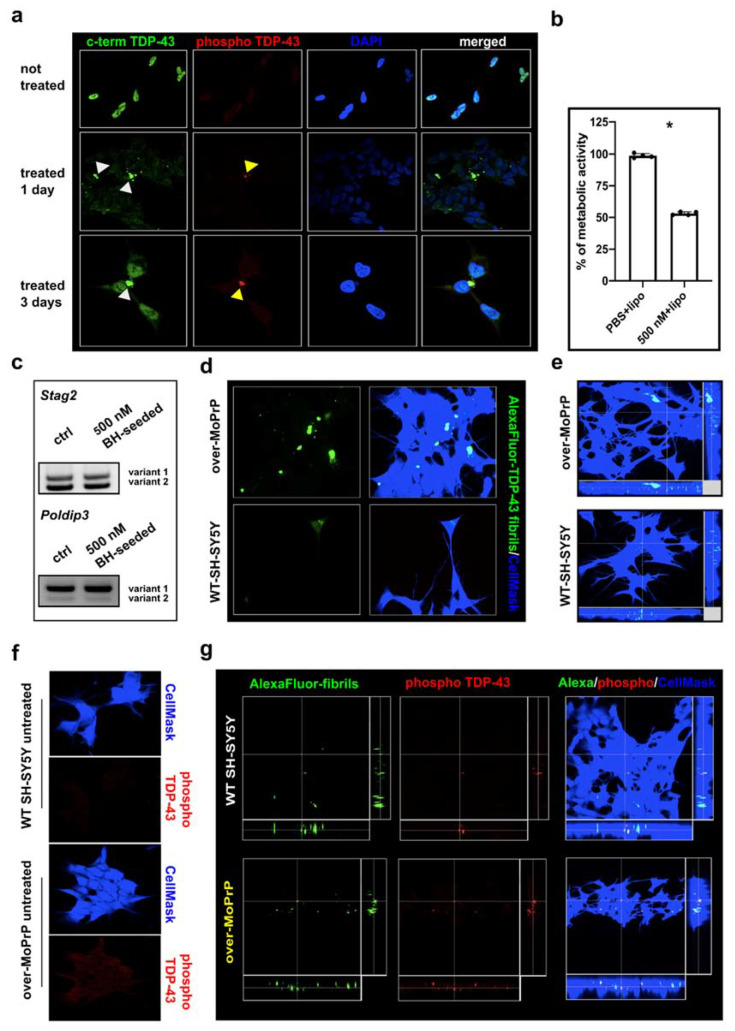
Internalization and phosphorylation of TDP-43 fibrils in WT and over-MoPrP SH-SY5Y cells (**a**) Representative immunofluorescence images of WT-SHSY5Y cells untreated and treated with 500 nM BH-seeded TDP-43 LCD sonicated fibrils mixed with lipofectamine to force their entrance within cells. Cells at 1 or 3 days post-treatment show the presence of green-fluorescent TDP-43 positive aggregates stained with the rabbit polyclonal anti-TDP-43 C-terminal antibody (white arrowheads). Several TDP-43 positive aggregates are recognized also by the anti-phospho TDP-43 (Ser409/Ser410) antibody (yellow arrowheads); (**b**) treatment with fibrils resulted in a reduction of cell vitality evaluated with the MTT assay. Cell vitality was evaluated as the percentage of treated cells with an active metabolism at the MTT assay compared to control cells treated with 1X PBS (corresponding to the 100% value of metabolic activity). Data in the graphs are reported as mean ±SD; each dot represents the mean value for the six replicates of each independent experiment; data were evaluated with Mann-Whitney test; * *p* < 0.05; (**c**) representative image of agarose gel showing that STAG2 and POLDIP3 genes splicing variant representation was not modified in SH-SY5Y cells upon treatment with 500 nM BH-seeded TDP-43 LCD sonicated fibrils. This analysis was performed in three independent experiments; (**d**) representative immunofluorescence images of WT and over-MoPrP SH-SY5Y cells treated with 1 μM AlexaFluor-488-conjugated TDP-43 LCD unseeded fibrils following 6 h of incubation and 72 h before analysis and (**e**) orthogonal views of the 3D Z-stack for WT and over-MoPrP cells exposed to the same treatment. The green staining corresponds to AlexaFluor-488-conjugated TDP-43 fibrils while the blue staining corresponds to the CellMask which defines the whole-cell body; (**f**) representative immunofluorescence images of untreated WT and over-MoPrP SH-SY5Y cells; blue staining corresponds to the CellMask while the red staining corresponds to the anti-phospho TDP-43 (Ser409/Ser410) antibody; (**g**) representative orthogonal views of the 3D Z-stack for WT and over-MoPrP cells treated with 1 μM AlexaFluor-488-conjugated TDP-43 LCD unseeded fibrils following 6 h of incubation and 24 h before analysis and trypan blue quenching before immunofluorescence; green dots represent TDP-43 conjugated fibrils and red spots correspond to TDP-43 aggregates recognized by the anti-phospho TDP-43 (Ser409/Ser410) antibody; all immunofluorescence experiments were replicated at least three times.

**Figure 4 viruses-13-01625-f004:**
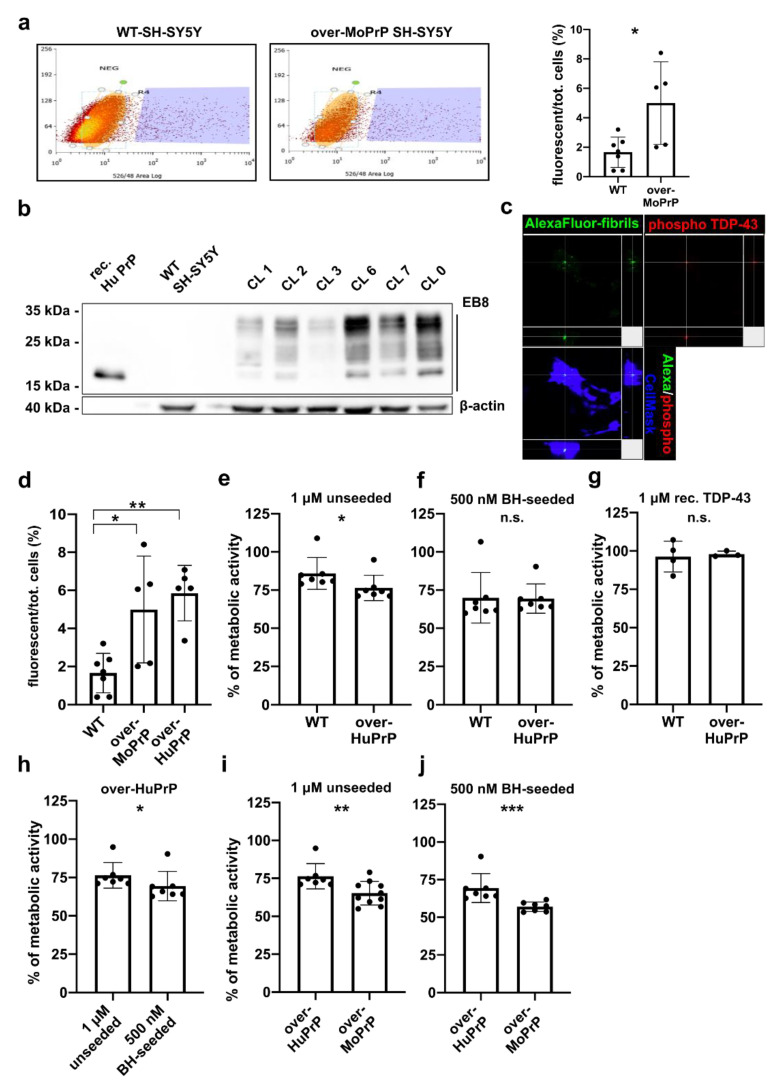
TDP-43 fibril uptake and toxicity in cells overexpressing mouse and human PrP^C^ (**a**) Cell sorter analysis of WT and over-MoPrP SH-SY5Y cells treated with 1 μM AlexaFluor-488-conjugated TDP-43 LCD unseeded fibrils following 6 h of incubation and 24 h before analysis. A control sample for each cell line was analyzed at every independent experiment to define the basal green fluorescence threshold. The graph on the right shows a significantly higher TDP-43 fibril uptake in over-MoPrP SH-SY5Y cells; data were evaluated with Mann-Whitney test; * *p* < 0.05; data in the graph are reported as mean ±SD; each dot represents an independent experiment; (**b**) generation of HuPrP^C^ overexpressing SH-SY5Y cells and selection of the most overexpressing clone (CL 6) by means of WB using the anti-PrP EB8 antibody; recombinant HuPrP (50 ng) was loaded as a positive control; after quantification 50 μg of total proteins were loaded for each cell lysate and β-actin was used as a loading control; (**c**) representative orthogonal views of the 3D Z-stack for over-HuPrP SH-SY5Y cells treated with 1 μM AlexaFluor-488-conjugated TDP-43 LCD unseeded fibrils following 6 h of incubation and 72 h before analysis; green dots represent TDP-43 conjugated fibrils and red spots correspond to TDP-43 aggregates recognized by the anti-phospho TDP-43 (Ser409/Ser410) antibody; (**d**) results of cell sorter analyses comparing green-fluorescent TDP-43 fibril internalization in WT, over-MoPrP and over-HuPrP SH-SY5Y cells; data were evaluated with Mann-Whitney test; * *p* < 0.05, ** *p* < 0.01; data in the graph are reported as mean ± SD; each dot represents an independent experiment; vitality of WT or over-HuPrP SH-SY5Y cells (evaluated as the percentage of cells with an active metabolism at the MTT assay) after treatment with (**e**) 1 μM of unseeded, (**f**) 500 nM BH-seeded TDP-43 LCD sonicated fibrils and (**g**) 1 μM recombinant monomeric TDP-43 LCD; (**h**) comparison of the toxic effect of unseeded and BH-seeded TDP-43 LCD fibrils on over-HuPrP SH-SY5Y cells; cell vitality of over-MoPrP and over-HuPrP SH-SY5Y cells after treatment with (**i**) 1 μM unseeded or (**j**) 500 nM BH-seeded TDP-43 LCD sonicated fibrils. Cell vitality was evaluated as the percentage of treated cells with an active metabolism at the MTT assay compared to control cells treated with 1X PBS (corresponding to the 100% value of metabolic activity). Data in the graphs are reported as mean ±SD; each dot represents the mean value for the six replicates of each independent experiment; data were evaluated with Mann-Whitney test; * *p* < 0.05, ** *p* < 0.01, *** *p* < 0.001, n.s.: not significant.

**Figure 5 viruses-13-01625-f005:**
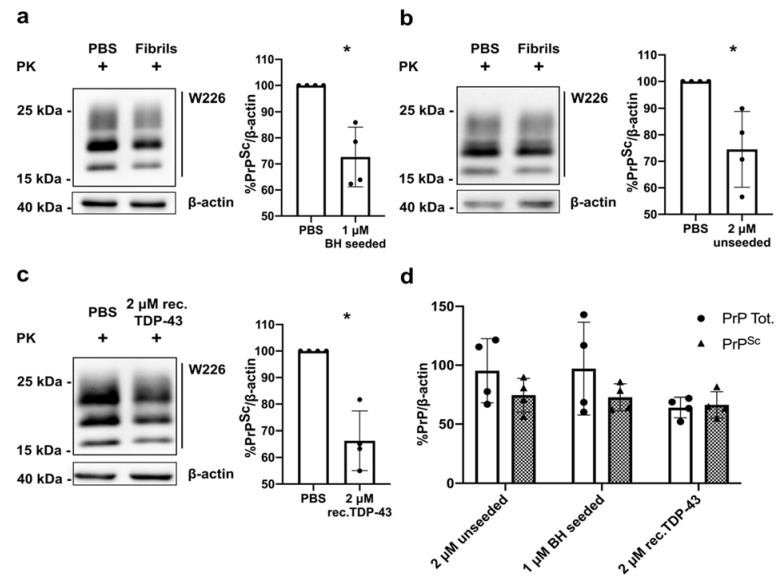
TDP-43-induced PrP^Sc^ clearance in ScN2a RML cells. Representative WB analysis of PK-resistant PrP^Sc^ and respective quantifications (graphs at the right side of each WB gel) of four independent experiments upon treatment of ScN2a RML cells with (**a**) 1 μM BH-seeded, (**b**) 2 μM unseeded TDP-43 LCD sonicated fibrils and (**c**) 2 μM recombinant monomeric TDP-43 LCD. Values are shown as percentage of the PK-resistant form PrP^Sc^ relative to β-actin, used as a loading control. Data were evaluated with Mann-Whitney test; * *p* < 0.05; data in the graph are reported as mean ± SD; each dot represents an independent experiment; (**d**) graphical representation of the quantification of PK-resistant and total PrP levels (i.e., the amount of PrP detected before PK treatment) for the four independent experiments upon treatment of ScN2a RML cells with 1 μM BH-seeded, 2 μM unseeded TDP-43 LCD sonicated fibrils and 2 μM recombinant monomeric TDP-43 LCD. Values are shown as a percentage relative to β-actin, used as a loading control.

## Data Availability

The data supporting the findings of this study are available from the corresponding author on request.

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
