# Peer review of "The Cellular Prion Protein Increases the Uptake and Toxicity of TDP-43 Fibrils"

_viruses, 2021, doi:10.3390/v13081625_

Round 1
Reviewer 1 Report
This study describes effects of different levels of PrPC expression in neuronal cell lines on the internalization and toxicity of TDP-43 fibrils. This work builds on previous studies showing roles of PrPC in mediating the uptake and toxicity of other aggregated proteins of significance in various neurodegenerative proteopathies. The findings are interesting, well performed and well described for the most part. However, I have some questions that should be addressed and suggestions for improvement.
- My understanding is that some prior related studies, e.g. with Aβ oligomers and fibrils, have concluded that PrPC interacts mostly with oligomers rather than fibrils. It would be of interest to discuss this issue and how it might relate to their current findings.
- The second paragraph in the Introduction is extensively and needlessly repetitive of the abstract and results sections.
- Figure 2: What do horizontal dotted lines in panels a-g indicate?
Also, a-g plots would better be plotted on 0-100% metabolic activity scale to give the reader a more accurate impression of the relative effects of the fibril treatments.
It would be helpful in the text and/or in the legend to indicate that the 100% value in this experiment is a mean from untreated cells (if that is the case). How variable are the mean MTT assay values for these controls? How different (statistically) are the MTT readings from fibril-treated cells from the controls?
- Were the TDP-43 preparations checked for endotoxin? This has been reported to bind to some amyloid fibrils, at least, and might thereby be concentrated for uptake into cells to a greater extent in fibril preparations compared to non-fibrillar preparations. The authors should consider the possibility that this might play a role in the greater toxicity of the fibril preparations.
- L382: suggest that “being the BH-seeded fibrils longer” be written as “…with the BH-seeded fibrils being longer…”
- L381-383: The conclusion that the unseeded and BH-seeded fibrils differ in morphology requires more documentation. Measurements of individual fibril widths in both preparations would be needed. It is not clear whether there might be a difference in fibril bundling rather than fibril width. Also, there appears to be more non-fibrillar material in the unseeded fibril preps that should be considered and discussed by as a having a potential effect on the cellular responses to these fibril preparations.
- L447-8: “Since trypsin action removed almost all fibrils located outside the cell membranes,…”. How is this known?
- L509: Suggest replacing “dispose” with “display”
- L572: Check spelling of “clearence”
- L587: Suggestion: “Despite” --> Although
11. L592: Suggestion: “knock-out” to knocked-out
Author Response
Please see the attachment for our point-by-point response to Reviewer's comments.

Reviewer 2 Report
In this very interesting study, Scialò and colleagues describe a new in vitro methodology to evaluate the interaction between recombinant PrP and TDP-43 and to generate in vitro-formed TDP-43 fibrils. These retain the characteristics of in vivo formed fibrils, such as the capacity of being internalized and phosphorylated by cells, and seem to exhibit, to a certain extent, the strain phenomenon, like prions and other prion-like proteins. The authors then elegantly go through a series of experiments that shed light on the role of PrP expression level and sequence on the uptake and toxicity of these TDP-43 fibrils.
The manuscript is brilliantly written and the figures are clear and beautiful. There are few things I can suggest to help improve the paper:
Figure 1c: if the labels are correctly positioned, the left panel is not very representative of what is stated in the text or shown in the right panel’s graph. In this particular blot the signal of the rec TDP-43 looks similar or a bit more intense than that of the TDP-43 fibrils. Maybe the image is incorrectly labelled and the two left-most bands corresponds to the fibril seeded reactions, as in Figure 1d left panel?
L356 and Figure S1b: in fact, cells express the TARDBP gen but it should not be stated that “detectable levels” of the protein are expressed without Western blot or Dot blot confirmation. Also, in figure S1b it seems weird to display Ct values in the vertical axis; the ΔCt or ΔΔCt values are usually represented instead.
Minor points:
L40: I think a comma is lacking?
L59: RT-QuIC stands for real-time quaking-induce conversion.
L323: recMoPrP and TDP-43 monomers or fibrils were mixed at a 1:50 molar ratio; I understand this means that 1 mol of TDP-43 was mixed with 50 moles of PrP, but it is a bit unclear.
L382: correct “the BH-seeded fibrils being longer…".
L397-399: excess of commas.
In the graphs displaying the % of metabolic activity of the cells in the vertical axis (Figures 2a-g, 3b, 4e-j), what does the dotted line at 70% represent?
Author Response

(The authors gave the same response as above.)

Round 2
Reviewer 1 Report
The authors have addressed my concerns adequately.